

# New insights into the eastern Subpolar North Atlantic meridional overturning circulation from OVIDE

Herlé Mercier[1], Damien Desbruyères[1], Pascale Lherminier[1], Antón Velo[2], Lidia Carracedo[1], Marcos Fontela[2,3], Fiz F. Pérez[2]

[1]University of Brest, CNRS, Ifremer, IRD, Laboratoire d'Océanographie Physique et Spatiale (LOPS), IUEM, Plouzané, 29280, France

[2]Instituto de Investigaciones Marinas (IIM), CSIC, Vigo, 36208, Spain

[3]Center of Marine Sciences (CCMAR), Universidade do Algarve, 8005-139 Faro, Portugal

*Correspondence to*: Herlé Mercier (Herle.Mercier@ifremer.fr)

**Abstract.** The Atlantic Meridional Overturning Circulation (AMOC) is a key component of the Earth's climate. However, there

are few long series of observations of the AMOC and the study of the mechanisms driving its variability depends mainly on numerical simulations. Here, we use four ocean circulation estimates produced by different data-driven approaches of increasing complexity to analyze the seasonal to decadal variability of the subpolar AMOC across the Greenland–Portugal OVIDE line since 1993. We show that the variance of the time series is dominated by seasonal variability, which is due to both seasonal variability in the volume of the AMOC limbs (linked to the seasonal cycle of density in the East Greenland Current) and to

seasonal variability in the transport of the Eastern Boundary Current. The decadal variability of the subpolar AMOC is mainly caused by changes in velocity, which after the mid-2000s are partly offset by changes in the volume of the AMOC limbs. This compensation means that the decadal variability of the AMOC is weaker and therefore more difficult to detect than the decadal variability of its velocity-driven and volume-driven components, which is highlighted by the formalism that we propose.

**Short summary for the general audience.** Here, we study the Atlantic Meridional Overturning Circulation (AMOC) measured between Greenland and Portugal between 1993–2021. We identify changes in AMOC limb volume and velocity as two major drivers of AMOC variability at subpolar latitudes. Volume variations dominate on the seasonal time scale, while velocity variations are more important on the decadal time scale. This decomposition proves useful for understanding the origin of the differences between AMOC time series.



## 1 Introduction


The Atlantic Meridional Overturning Circulation (AMOC) is key in the climate system through the uptake and redistribution of heat, freshwater, and dissolved inorganic carbon across latitudes in the Atlantic Ocean (e. g. Pérez et al. 2013; Bryden et. al., 2020; Williams et al., 2021; Messias and Mercier, 2022). Paleoclimatic evidence suggests that abrupt changes in North Atlantic climate occurred during glacial and interglacial periods, with transition periods of a few decades, and identifies AMOC as a key

feature associated with these abrupt changes (Lynch-Stieglitz, 2007). Today, as the climate is perturbed by human activity, climate projections suggest that the AMOC will decrease in response to anthropogenic forcing (Weijer et al., 2020). However, the magnitude and timing of this decline remains uncertain and it is still not known whether this decline has already begun. This critical role of the AMOC in climate change has highlighted the need to monitor its evolution under current anthropogenic forcing and has prompted unprecedented efforts over the past decades to establish AMOC observing systems (Srokosz and Bryden,

2015; Frajka-Williams et al., 2019; McCarthy et al., 2019).

In the subtropical North Atlantic, the trans-Atlantic RAPID network (26.5°N), deployed since 2004, has shown variability in the MOC from weeks to decades (Srokosz and Bryden, 2015; we use the acronym MOC to designate a measure of the AMOC at a specific location), consistent with that of the Meridional Overturning Variability Experiment (MOVE) network at 16°N (Jackson

et al., 2022). A notable feature in this time series is that the MOC has shifted to a reduced circulation state since 2008 (Smeed et al., 2018) with hints of a potential recovery in recent years (Moat et al, 2020). A second signal of interest is a wind-forced sharp decrease in the amplitude of the MOC for several months in 2009 that created a heat transport anomaly partly responsible for the decrease in the heat content of the subpolar gyre half a decade later (Bryden et al., 2020). Using a proxy method which makes it possible to study a longer period, Worthington et al. (2021) concluded that there has been no decline in MOC at 26.5°N since

the early 1980s. At seasonal to interannual time scales, MOC variability at 26.5°N has been shown to be related to the variability in the wind stress curl through variability of the Ekman transport and mediation by Rossby waves (Zhao and Johns 2014a, b; Kansow et al. 2010).

In the subpolar latitudes of the North Atlantic, the Overturning in the Subpolar North Atlantic Program (OSNAP) network, which

covers the Labrador Sea, the Irminger Sea and the Iceland Basin to the Scottish Shelf, operates since 2014. OSNAP has also revealed significant variability for all resolved frequencies (Li et al., 2021). The MOC time series at OVIDE (Observatoire de la Variabilité Interannuelle à DÉcennale), which used hydrography and altimetry to reconstruct the MOC between Greenland and Portugal since 1993 (Mercier et al., 2015; Frakja-Williams, 2019; Figure 1 for section location), also shows strong variability at all resolved time scales. OSNAP has highlighted the dominant role of the eastern subpolar gyre in shaping the mean state and

the variability of the MOC, most of the subpolar overturning occurring between Greenland and Scotland (Lozier et al. 2019; Li et al., 2021). The low-frequency variability of the MOC in the subpolar gyre has been linked to the variability of buoyancy fluxes to the north of the observation sections on multi-year time scales (Desbruyères et al., 2019), with storage becoming important on shorter time scales (Petit et al., 2020). At intra-annual frequencies, MOC seasonality at OSNAP can be explained by both the seasonality in the water mass transformation and the seasonality in the Ekman transport (Fu et al., 2023). Considering OSNAP-

East (Figure 1), Wang et al. (2021) established a link between MOC seasonality and seasonal displacement in the Irminger Basin of $\sigma_{moc}$, the isopycnal separating the MOC upper limb from the lower limb. Tooth et al. (2023) concluded that seasonality in the upper East Greenland Current (EGC) transport must also be considered to explain the full seasonality of the MOC across OSNAP-East. A remarkable feature of the North Atlantic subpolar gyre is its deep convection sites in the Labrador Sea, southeast Greenland and Irminger Sea, subject to significant interannual to decadal variations in the properties (e.g. density) and volumes

of the water masses formed (Yashayaev and Loder, 2016; Piron et al., 2016; de Jong et al., 2018; Zunino et al. 2020). This





variability in the density of the 0–1000 m layer in the Irminger Sea has been shown to be a key player in setting the MOC strength on interannual to decadal timescales (Chafik et al., 2022).

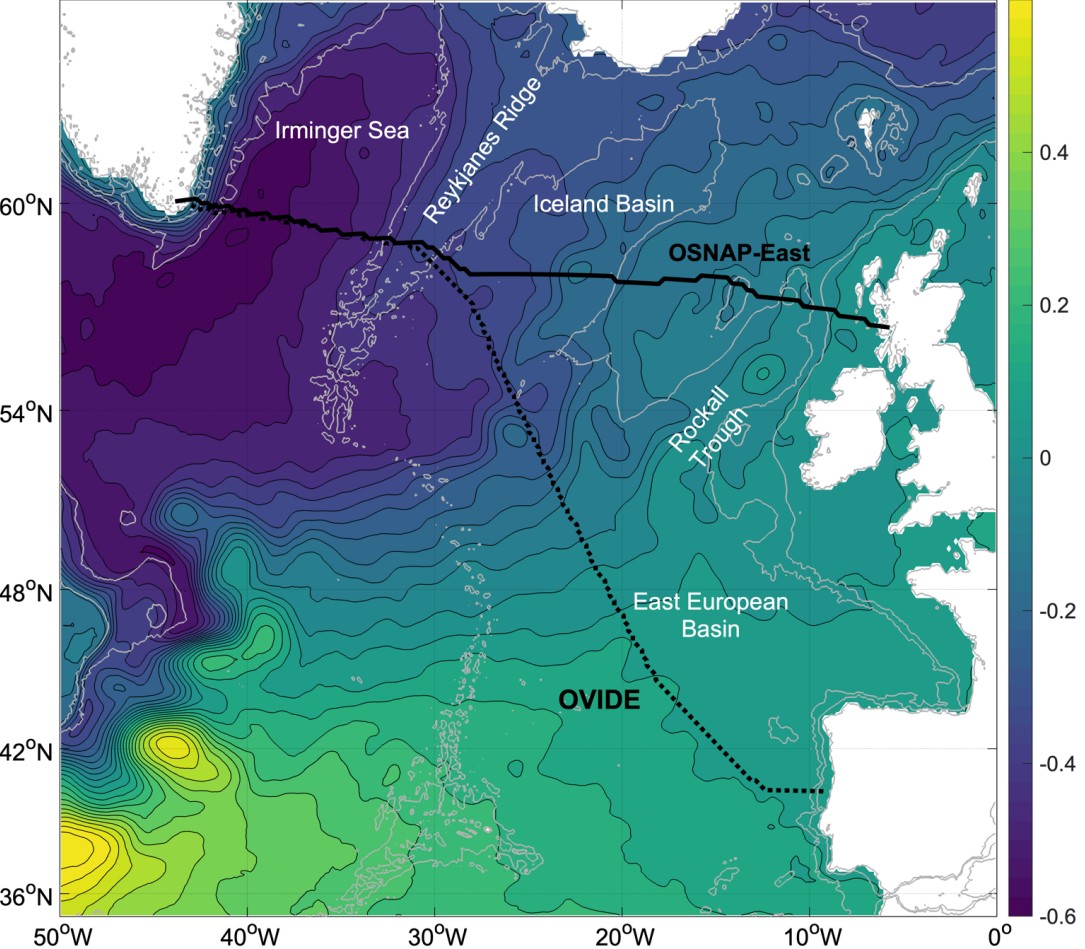

**Figure 1: OVIDE and OSNAP East lines plotted over the mean sea surface height (cm) from AVISO. The 200 and 2000 meters isobaths are plotted in grey.**

The relationship between subpolar and subtropical latitude AMOC is the subject of a large number of studies. Among them, some have linked density variations in the central Labrador Sea to the strength of the AMOC at subtropical latitudes (e.g. Böning et al., 2006; Robson et al., 2014; Yeager et al., 2021). These density anomalies are thought to spread southwards from the Labrador Sea along the deep western boundary current or by interior routes (e.g. Zhang et al., 2010), thus modifying the zonal density gradient and consequently the AMOC. Intriguingly, observations at 26.5°N show that most of the variability observed in the deep western boundary current does not occur at the level of the Labrador Sea Water (LSW) but below it in the Lower North Atlantic Deep Water (LNADW), a water mass originating from the Nordic Seas (McCarthy et al., 2012; Smeed et al., 2014; Zou et al., 2018; Johns et al., 2023). In a recent numerical study, Kostov et al. (2023) proposed mechanisms, activated by density anomalies at the south-western boundary of the Labrador Sea, by which the North Atlantic Current (NAC), and thus the eastern subpolar gyre, plays a central role in linking Labrador Sea surface density anomalies to LNADW variability at RAPID in about half a decade. On decadal time scales, Desbruyères et al. (2013) showed that the decadal variability of the MOC across



90 the OVIDE line is associated with synchronized changes in the NAC subpolar and subtropical components, the latter being the main source of variability and providing a link between subtropical and subpolar variability. Nevertheless, the fact that the drivers of AMOC variability depend on the latitude at which the AMOC is studied (e.g. Kostov et al., 2021) complicates the identification in the observations of a connection between the subpolar AMOC and the subtropical AMOC. To achieve this latter goal, sustained networks of observations to better understand the variability of the AMOC and its components as well as synergies with modeling and theoretical studies must be pursued. Here, we study the variability of the eastern subpolar AMOC

95 and its link with the variability of its components.

The aim of this article is to analyse MOC strength time series across the OVIDE line between 1993 and 2021 in terms of seasonal to decadal variability. This work is based on four time series whose common feature is that they were derived using data-driven approaches of varying complexity, which complements recent studies based on prognostic numerical simulations. We show that

100 the variability of the MOC at OVIDE can be effectively decomposed into a volume-driven term, linked to changes in the volume of the MOC branches at constant velocity, and a velocity-driven term at constant volume, which sheds light on the mechanisms of the observed seasonal to decadal variability. The data are presented in section 2, the methodology in section 3, the results in section 4. We end the paper with a discussion in section 5 and concluding remarks in section 6.

## 2 Data

105 ### 2.1 OVIDE hydrographic data

We used the Greenland to Portugal OVIDE hydrographic line, referred to as A25 by GO-SHIP (Global Ocean Ship-based Hydrographic Investigations Program, Sloyan et al., 2019), which was occupied every second year between 2002 and 2018 and repeated again in 2021 (Figure 1). The surveys last about three weeks and have always been carried out between May and July. The FOUREX hydrographic line carried out in August–September 1997 along a nearby track was also used. Each section

110 comprises at least 92 hydrographic stations with a nominal station spacing of 25 NM reduced to 10 NM or less over continental slopes and oceanic ridges with the exception of the 2014 occupation that, due to limited ship-time and repeated stations for sampling GEOTRACES program core parameters, had coarser station spacing away from fronts and boundary currents (Lherminier et al., 2007; Lherminier et al., 2010; Gourcuff et al., 2011; Mercier et al., 2015; Zunino et al., 2017). FOUREX and OVIDE temperature, pressure, and conductivity measurement accuracies meet the GO-SHIP requirements (Sloyan et al., 2019).

115 ### 2.2 Objective analyses of temperature and salinity measurements

Coriolis Ocean dataset for ReAnalysis (CORA) is a global objective analysis of delayed-mode quality controlled in situ temperature and salinity profiles (Szekely et al., 2019). Here, we used CORA v5.2 monthly gridded fields from 1993 to 2021. CORA v5.2 grid horizontal resolution is 0.5° x cosine(latitude) in latitude and 0.5° in longitude, and it has 152 irregularly spaced levels. Maximum analysis depth is 2000 m.


EN4 is a collection of objective analyses of potential temperature and salinity profiles (Good et al., 2013). Here, we used EN4 version 4.2.2 monthly gridded fields with the bias corrections from Cheng et al. (2014) for the time period from 1993 through 2021. EN4 provides gridded fields of potential temperature and salinity with associated errors, with a monthly temporal resolution, 1° by 1° horizontal resolution and 42 irregularly spaced depth levels.


We defined a grid along the path of the OVIDE line, referred herein as the OVIDE line grid, with a horizontal resolution of 7 km and vertical resolution of 1 m. The CORA v5.2 and EN4 temperature and salinity fields were linearly interpolated to the locations of the OVIDE line grid.



### 2.3 GloSea5 reanalysis

GloSea5 is an ocean reanalysis based on the ensemble prediction system built around the high-resolution version of the Met Office climate prediction model: HadGEM3 family atmosphere-ocean coupled climate model (MacLachlan et al., 2015; Scaife et al., 2014). The reanalysis uses most of available satellite and in situ data and an incremental three-dimensional variational first guess at appropriate time (FGAT) data assimilation system (Jackson et al., 2016; MacLachlan et al., 2015). Increments are applied to temperature and salinity fields. The ocean general circulation model is the Nucleus for European Modelling of the

Ocean (NEMO) model in its ORCA0.25 configuration (0.25° horizontal resolution with 75 irregularly spaced vertical levels). GloSea5 reanalysis is distributed by Copernicus Marine Environment Monitoring Service (CMEMS) in interpolated form on a grid common to other reanalyses. For better accuracy of transport determination, here we used the monthly fields of potential temperature, salinity and velocity from GloSea5 on the ORCA025 native grid along the OVIDE line, provided by L. Jackson (personal communication, 2023) for 1993–2021.

**2.4 State estimate**

ECCO is a state estimate that combines the MIT general circulation model and most of available satellite and in situ data to produce a physically consistent estimate of the global ocean using an adjoint-based four-dimensional data assimilation system which optimizes the solution through adjusting initial conditions and parameters (including surface fluxes, wind stresses, and mixing parameters) (Fukumori et al., 2018). Here, we used monthly-averaged potential temperature, salinity and velocity fields

from ECCO V4r4 on the native grid that covers the time period from 1992 through 2017 and has a resolution of 1° in the horizontal and 50 irregularly spaced vertical levels. The reader is referred to Jackson et al. (2016 and 2019) for a discussion of North Atlantic circulation features from GloSys5 and ECCO in perspective with estimation methods and other analyses.

### 2.5 Sea surface height

The daily altimeter sea surface height data from the Merged Absolute Dynamic Topography of Ssalto/Duacs AVISO (Archiving,

Validation and Interpretation of Satellite Oceanographic data center) distributed by CMEMS on a 1/3° grid were interpolated on the OVIDE line grid. We used the monthly surface geostrophic velocities perpendicular to the OVIDE line that were computed for 1993–2021 from these sea surface heights.

### 2.6 NCEP atmospheric reanalysis

The 6-hourly wind stress data of the global atmospheric National Centers for Environmental Prediction (NCEP) / National Center

for Atmospheric Research (NCAR) reanalysis (Kalnay et al., 1996) were linearly interpolated to the locations of the OVIDE line. We used monthly Ekman transports perpendicular to the OVIDE line that were then calculated for the time period 1993–2021.

### 3 Methods

### 3.1 Determination of absolute velocities for OVIDE hydrographic lines, CORA and EN4

For each occupation of the OVIDE hydrographic line over the period 2002-2021 and for the 1997 FOUREX line, the MOC was calculated using an inverse model constrained by volume conservation. Geostrophic velocities are obtained by combining geostrophic shears calculated from temperature and salinity observations measured at hydrographic stations with currents measured at the time of the cruise by SADCP (Ship-mounted Acoustic Current Profilers). The Ekman transport estimated from NCEP is included in a surface layer (0–30 m). The inverse model calculates a correction to be applied to each pair of hydrographic



stations to satisfy volume conservation (Lherminier et al., 2007; Lherminier et al., 2010; Gourcuff et al., 2011; Mercier et al., 2015; Zunino et al., 2017).

For the objective mappings CORA and EN4, the 0–2000 m geostrophic velocity was first computed at any point of the OVIDE line grid by combining the surface-referenced geostrophic current shears computed from CORA v5.2 or EN4 4.2.2 fields and
surface geostrophic velocities obtained from altimetry. An Ekman velocity equal to the Ekman transport calculated from NCEP divided by the thickness of the shallower vertical layer was then added to the geostrophic velocity at this level. The method follows that of Mercier et al (2015) which can be referred to for further details.

### 3.2 AMOC estimation

In the North Atlantic subpolar gyre, the water mass transformation from the upper to the lower branch of the AMOC takes place
through progressive cooling of the winter mixed layer along the cyclonic subpolar gyre circulation, so that the upper and lower branches of the AMOC overlap in depth. The AMOC strength must therefore be calculated in density coordinates to capture all the associated water mass conversion (Lherminier et al., 2007; Mercier et al., 2015; Lozier et al., 2019). The AMOC in density coordinate across the OVIDE line $\psi(t)$ reads:

$$\psi(t) = \int_{surface}^{\sigma_{moc}} \int_{Portugal}^{Greenland} v(x, \sigma, t)\, dx\, d\sigma, \qquad (1)$$

where $\sigma$ is potential density referred to 1000 db, $x$ is along-section distance, $t$ is time, $\sigma_{moc}(t)$ is the density at the maximum of the AMOC stream function $\psi(\sigma, t)$. $\sigma_{moc}$ defines the isopycnal that separates the upper and lower limbs of the AMOC. $v(\sigma, x, t)$ is the gridded velocity field perpendicular to the OVIDE line grid. $\psi(t)$ was calculated for all datasets, the OVIDE
hydrographic sections ($\psi_{hydro}$), the CORA and EN4 objective analyses ($\psi_{cora}$, $\psi_{en4}$), the GloSea5 reanalysis ($\psi_{glosea5}$) and the ECCO state estimate ($\psi_{ecco}$). In the following, we will use the term "analyses" to refer to all these products, without singling out any one in particular.

The AMOC strength was calculated by integrating $\psi(\sigma, t)$ from surface down to its maximum. The reason for not integrating
from the bottom as usually done (Lherminier et al., 2007) is that CORA is only available for 0–2000 m, and integration from the surface means that we can use the same method for all analyses. The Arctic volume budget imposes a net northward transport across the OVIDE line which has been estimated at 0.8 Sv (1 Sv = $10^6$ m³ s⁻¹) on average from OVIDE inversions (Mercier et al., 2015). Here, positive transports are directed northward. This transport occurs in the upper branch of the AMOC that therefore has a strength that is greater than that of the lower branch by the value of this net transport. A net northward transport is also
present in ECCO and GloSea5 with a value of 0.2 and 0.8 Sv respectively (Fig. S1).

### 3.3 AMOC decomposition

The aim of this section is to propose a decomposition of the AMOC strength that decouples the time variations in AMOC strength due, on the one hand, to changes in the volume of the upper layer of the AMOC due to the variability in $\sigma_{moc}$ and, on the other hand, to changes in velocity. We start by decomposing the AMOC $\psi(t)$ into a time-averaged component $\bar{\psi}$ and a time-
dependent component $\psi'(t)$ where the overbar denotes the time mean and the prime the variability about the time mean, following Desbruyères et al. (2013). For that purpose, we expand $v(\sigma, x, t)$ as:

$$v(z, x, t) = \bar{v}(z, x) + v'(z, x, t), \qquad (2)$$



and define

$$\bar{\psi} = \int_0^{\overline{\sigma_{\text{moc}}}} \int_{Greenland}^{Portugal} \bar{v}(\sigma, x) \, dx \, d\sigma \tag{3}$$

where $\overline{\sigma_{\text{moc}}}$ is the time-averaged density of the maximum of the overturning stream function.


$\psi(t)$ can be written as well as:

$$\psi(t) = \int_0^{Z\sigma_{moc}} \int_{Portugal}^{Greenland} v(x, z, t) \, dx \, dz, \tag{4}$$

where z is depth and $Z\sigma_{moc}(x, t)$ is the depth of $\sigma_{moc}$, computed using the monthly density field along the section. It follows that the time averaged AMOC can be rewritten as

$$\bar{\psi} = \int_0^{\overline{Z\sigma_{\text{moc}}}} \int_{Greenland}^{Portugal} \bar{v}(z, x) \, dx \, dz, \tag{5}$$

where $\overline{Z\sigma_{\text{moc}}}$ is computed using the time-averaged density field along the section. It follows that

$$\psi'(t) = \int_0^{\overline{Z\sigma_{\text{moc}}}} \int_{Greenland}^{Portugal} v'(z, x, t) \, dx \, dz + \int_{\overline{Z\sigma_{\text{moc}}}}^{Z\sigma_{moc}} \int_{Greenland}^{Portugal} \bar{v}(z, x, t) \, dx \, dz \ ...$$

$$+ \int_{\overline{Z\sigma_{\text{moc}}}}^{Z\sigma_{moc}} \int_{Greenland}^{Portugal} v'(z, x, t) \, dx \, dz, \tag{6}$$


or

$$\psi'(t) = \psi'_v(t) + \psi'_{\sigma_{moc}}(t) + \psi'_{v\,\sigma_{moc}}(t). \tag{7}$$

$\psi'_v$ is the contribution to the AMOC variability due to the time variability of the velocity field in the density layer bounded by the constant isopycnal limit $\overline{\sigma_{\text{moc}}}$. $\psi'_{\sigma_{moc}}$ is the contribution to the AMOC variability due to the time change in the lower limit in density of the AMOC upper limb or, in other words, to the change in volume of the upper limb of the AMOC acting on the mean velocity field. A change in volume will be all the more effective in producing an AMOC strength anomaly if it occurs in a region where the mean current is strong (e.g. NAC or EGC). $\psi'_{v\,\sigma_{moc}}$ is the variability due to the correlation between the
velocity fluctuations and the fluctuations in the depth of $\sigma_{moc}$. Note that $\psi_{Ekman}(t)$, the Ekman transport perpendicular to the section and integrated from coast to coast along the section, is included in the velocity component. The decomposition in Eq. 7 is exact, and we have verified that in our computations the sum of the three terms plus the mean is strictly equal to the AMOC time series.

**3.4 Determination of seasonal cycle and statistics**
The statistics were calculated by considering the time series as a series of N correlated samples whose effective number of degrees of freedom is given by N/2τ where τ is the integral time defined from the auto-correlation function of the time series calculated after subtracting the non-random components which are the mean, the trend and the average seasonal cycle of the series (Thomson and Emery, 2014). The confidence interval on the cross-correlation coefficient r between two time series a and



b was calculated by noting that ln[(1+r)/(1-r)] is a Gaussian random variable (see Thomson and Emery, 2014). The effective number of degrees of freedom was calculated in this case following Bretherton et al. (1999) and is given by $N(1-r_a r_b)/(1+r_a r_b)$ where $r_a$ and $r_b$ are the values of the lag 1 auto-correlations of the a and b series.

Time series trends were determined by least-squares fitting of a first-order polynomial to the observations. The trend error was
determined as the standard deviation of a set of 2,000 trend estimates calculated from perturbed time series obtained by randomly permuting blocks of least-squares adjustment residuals. This method, known as moving block bootstrap (MBB) resampling (Mudelsee, 2019), uses blocks of residuals whose length depends on the temporal correlation between the residuals and which therefore preserve the correlation between the residuals during permutation.

The seasonal cycle of the time series was obtained by removing the trend, then applying a high-pass filter with a cut-off frequency of two years, then averaging the high-pass filtered time series variable for each calendar month separately.  The standard error was calculated on the assumption that AMOC observations for a given month one year apart are independent.

**4 Results**

**4.1 MOC time series**

Time series of MOC strength $\psi(t)$ show for all the analyses an energetic seasonality as well as a smaller but discernible inter-annual to decadal variability (Figure 2). Over the period 1993-2021, the mean values of $\psi_{cora}$ (19.7±0.4 Sv), $\psi_{en4}$ (20.3±0.4 Sv) and $\psi_{glosea5}$ (20.0±0.3 Sv) are close with overlapping standard errors (Table 1). $\psi_{cora}$ and $\psi_{en4}$ have a small bias with respect to $\psi_{hydro}$, the MOC strengths obtained by analysis of the OVIDE hydrographic lines (-0.23 and 0.23 Sv, respectively, Table 1). These biases are for the June-July period, when the cruises were carried out, with the sole exception of the 1997 cruise
carried out in September. $\psi_{glosea5}$ overestimates the MOC strength compared with $\psi_{hydro}$ by 2.75 Sv. $\psi_{cora}$ and $\psi_{en4}$ show very similar signals with the exception of certain winters with marked differences in MOC strength (e.g. 2014). These two analyses are positively correlated (r = 0.63, Table 2). $\psi_{glosea5}$ shows correlations of 0.22 and 0.17 with $\psi_{cora}$ and $\psi_{en4}$, respectively (Table 2). Although significantly different from zero at the 99% confidence level, these correlations are weaker than those between $\psi_{cora}$ and $\psi_{en4}$ and reflect, despite broadly similar multi-year variability, differences at intra-annual frequencies
(e.g. 2006 to 2009, Figure 2). The mean value of $\psi_{ecco}$ (15.2±0.4 Sv) is significantly lower than those of $\psi_{cora}$, $\psi_{en4}$ and $\psi_{glosea5}$. This is not primarily due to the different periods under consideration, as for the period covered by ECCO $\psi_{ecco}$ is lower than $\psi_{hydro}$ by -2.95 Sv on average (Table 1). In brief, the analyses show a bias with hydrography that is positive for GloSea5, and negative for ECCO and a good agreement between hydrography, EN4 and CORA. It should be noted that the various AMOC strength estimates are not independent, as they are based on largely similar data sets (altimetry, hydrographic profiles). However,
none of them assimilates the S-ADCP data that are decisive in estimating $\psi_{hydro}$ (Lherminier et al., 2007).

The spatial evolution of the cumulated transport of the upper limb of the MOC from Greenland to Portugal, averaged over the duration of the hydrographic cruises, provides a better understanding of the origin of the biases observed in the restitution of the MOC by the different analyses (Figure 3). GloSea5 has a larger bias with $\psi_{hydro}$, but shows the closest match to hydrographical
observation regarding the along-section transport distribution, except for the presence of a ~2 Sv northward eastern boundary current, which at that time of the year is not present in the hydrography, the objective analyses or the state estimates. Overall, there is a good agreement between CORA, EN4 and the hydrography, even though the objective analyses show a more intense NAC transport than that observed during the cruises. The ECCO state estimate shows a lower NAC transport than in the observations, which is the main cause of a lower AMOC strength compared to the hydrography.


**Figure 2: Time series of MOC strength at OVIDE from CORA (cyan), EN4 (green) and GloSea5 (orange) and ECCO (blue) in Sv. MOC strengths and associated standard errors estimated from OVIDE hydrographic lines are plotted in red. The annual means of each time series (black, horizontal lines) are also indicated.**




|  | <MOC > | std(MOC) | std($MOC_{intra}$) | rms(MOC) with hydro | <$\sigma_{moc}$ > | std($\sigma_{moc}$) | rms($\sigma_{moc}$) with hydro |
|---|---|---|---|---|---|---|---|
| CORA | 19.7±0.4 | 4.3 | 4.0 | 2.6 (-0.23) | 32.18 | 0.07 | 0.06 (0.02) |
| EN4 | 20.3±0.4 | 4.3 | 3.7 | 2.2 (0.23) | 32.19 | 0.07 | 0.06 (0.04) |
| GloSea5 | 20.0±0.3 | 2.7 | 2.3 | 2.4 (2.75) | 32.17 | 0.08 | 0.05 (0.04) |
| ECCO | 15.2±0.4 | 2.5 | 2.0 | 2.3 (-2.95) | 32.16 | 0.08 | 0.04 (0.01) |
| hydro | 17.4±0.6 | 2.1 |  | 0 (0) | 32.15 | 0.03 | 0 (0) |

**Table 1: Statistics for MOC strength time series in Sv and $\sigma_{moc}$ time series in kg m$^{-3}$: Mean (<•>), standard deviation (std(•)), and root mean square differences with estimates from OVIDE hydrography rms(•). The standard error is reported for <MOC>. Biases with hydrography are reported in parentheses in the rms columns. MOC stands for MOC strength, $MOC_{intra}$ for the intra-annual component of the MOC strength and hydro refers to estimates from OVIDE repeated hydrography.**

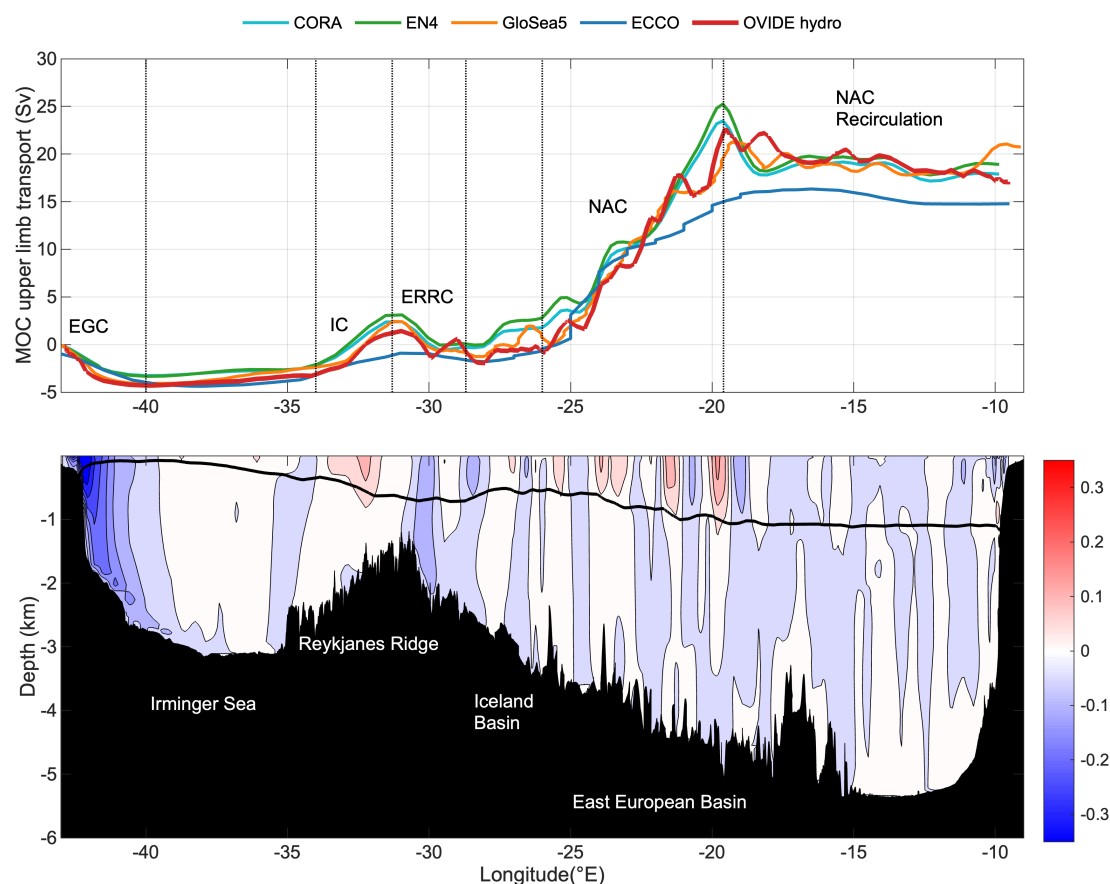

**Figure 3: (upper panel) Cumulated transport from Greenland to Portugal along the OVIDE line in the MOC upper limb in Sv for CORA (cyan), EN4 (green), GloSea5 (orange) and ECCO (blue) averaged over the time period covered by the cruises, and the average of the 1997–2018 OVIDE hydrographic line (red). (lower panel) geostrophic velocity (m s$^{-1}$) perpendicular to the OVIDE line averaged over the OVIDE cruises and adapted from Daniault et al. (2016). Positive velocities are directed northwards. The continuous black line is $\sigma_{moc}$ averaged over the OVIDE cruises. Main bathymetric features are indicated. EGC stands for East Greenland Current, IC for Irminger Current, ERRC for East Reykjanes Ridge Current, NAC for North Atlantic Current, with the dotted vertical lines indicating the extension in longitude of the currents.**




**Figure 4 : Times series of potential density reference to 1000 dbar at the maximum of the overturning stream function ($\sigma_{moc}$) in kg m$^-$$^3$ for CORA (cyan), EN4 (green) and GloSea5 (orange) and ECCO (blue). $\sigma_{moc}$ from OVIDE hydrographic lines are plotted as red dots.**




The MOC annual average in Figure 2 shows a decreasing MOC in the 1990s and early 2000s for $\psi_{cora}$, $\psi_{en4}$ and $\psi_{glosea5}$, more
pronounced for $\psi_{en4}$. For $\psi_{cora}$ and $\psi_{glosea5}$, this is followed by a period of lower MOC between 2000 and 2008 and a period
of MOC fluctuations around a higher mean until 2021. $\psi_{en4}$ fluctuates around a high MOC value from 2003 onwards. The annual
mean from $\psi_{ecco}$ shows a decrease in MOC from the early 2000s to the late 2010s, with inter-annual variability superimposed
(Figure 2d). In the late 2010's, the average annual of $\psi_{ecco}$ is below 15 Sv. It is over this last period that the negative bias in

regard to $\psi_{hydro}$ is most noticeable. Trends were calculated for the different time series over their entire duration, but none were
significant at the 95% confidence level and they are not described here. The magnitude of MOC strength variability is measured
by the standard deviation of the time series (Table 1). With a standard deviation equal to 4.3 Sv, $\psi_{cora}$ and $\psi_{en4}$ show
significantly more variability than $\psi_{glosea5}$ (standard deviation of 2.7 Sv) and $\psi_{ecco}$ (2.5 Sv) (Table 1). For each analysis, the
intra-annual component largely explains the intensity of the variability (Table 1). The differences between the standard deviations

are therefore mainly due to differences in seasonal signal amplitude.

Time series of $\sigma_{moc}$ show strong intra-annual variability superimposed on interannual to decadal variability (Figure 4). During
an intra-annual cycle, $\sigma_{moc}$ is at its densest at the end of winter. $\sigma_{moc}$ has an average value of between 32.16 and 32.19 kg m$^{-3}$
depending on the analysis considered (Table1). All the analyses show that the densest value of $\sigma_{moc} \sim 32.25$ kg m$^{-3}$ occurred in

the early 1990s. Then, $\sigma_{moc}$ gradually decreased to reach 32.1 kg m$^{-3}$ in the mid-2000s to become again denser $\sim 32.2$ kg m$^{-3}$ in
2015-2016. $\sigma_{moc}$ follows the trends of the subpolar gyre, which became less dense (warming) between the mid-1990s and 2006,
then became denser (cooling) until 2016, and has been warming again since then (Desbruyères et al., 2015, 2021). Higher
frequency inter-annual variability is superimposed on this decadal cycle.

|          | EN4  | GloSea5 | ECCO |
|----------|------|---------|------|
| CORA     | 0.63 | 0.22    | 0.34 |
| EN4      |      | 0.17    | 0.23 |
| GloSea5  |      |         | 0.28 |

**Table 2: Cross-covariances of MOC time series in Sv (all reported cross-covariances are significantly different from zero at the 99% confidence level).**

|         | $std(\psi'_v)$ | $std(\psi'_{\sigma_{moc}})$ | $std(\psi'_{v\,\sigma_{moc}})$ | $std(\psi'_{Ekman})$ |
|---------|----------------|------------------------------|--------------------------------|----------------------|
| CORA    | 2.56           | 3.17                         | 0.92                           |                      |
| EN4     | 2.70           | 3.33                         | 0.94                           |                      |
| GloSea5 | 2.12           | 2.21                         | 0.91                           |                      |
| ECCO    | 1.18           | 1.94                         | 0.42                           |                      |
| NCEP    |                |                              |                                | 1.40                 |

**Table 3: Standard deviations of MOC decomposition terms (Eq. 7 and following discussion) reported in Sv.**

**4.2 MOC decomposition**

We decomposed the MOC strength time series according to equation 7. $\psi'_v$ and $\psi'_{\sigma_{moc}}$ for the four analyses are shown in Figure
5 and 6 respectively; the associated statistics are presented in Table 3. The two components $\psi'_v$ and $\psi'_{\sigma_{moc}}$ contribute to the
interannual to decadal MOC variability, whereas $\psi'_{\sigma_{moc}}$ explains most of the seasonality of the MOC strength, which is
confirmed by a spectral analysis (Figure S2). The variance due to velocity fluctuations amounts to 37 % (ECCO), 65.2 %

(CORA), 66.9 % (EN4) and 92% (GloSea5) of the variance due to $\sigma_{moc}$ depth variations. Variability in the depth of $\sigma_{moc}$ is
therefore the dominant mechanism for generating variance in the subpolar MOC. It is worth noting that, as we shall see below,





this result is representative of the seasonal scale, which is the time scale that predominates in the variability spectrum, but not of the inter-annual to decadal scales. The contribution of the coupled term $\psi'_{v\,\sigma_{moc}}$, which represents the correlation between

**Figure 5 :** Coloured solid lines are times series of $\psi'_v$ in Sv for CORA (cyan), EN4 (green), GloSea5 (orange) and ECCO (blue). The black solid lines are $\psi'$. The coloured (black) dashed lines are $\psi'_v$ ($\psi'$) after low-pass filtering with a moving average of 60 months.



**Figure 6 :** Coloured solid lines are times series of $\psi'_{\sigma_{moc}}$ in Sv for CORA (cyan), EN4 (green), GloSea5 (orange) and ECCO (blue). The black solid lines are $\psi'$. The coloured (black) dashed lines are $\psi'_{\sigma_{moc}}$ ($\psi'$) after low-pass filtering with a moving average of 60 months.




velocity fluctuations and $\sigma_{moc}$ depth fluctuations, explains less than 10% of the variance. The variance of Ekman transport is included in $\psi'_v$ and accounts for 18% (EN4), 19.5% (CORA), 40% (GloSea5) and 52.1% (ECCO) of the variance of $\psi'_{\sigma_{moc}}$. Note that these percentage variations therefore only reflect variations in the amplitude of $\psi'_{\sigma_{moc}}$ as here we estimate $\psi'_{Ekman}$ from NCEP regardless of the analysis considered. The variability of $\psi'_{Ekman}$ and $\psi'_{v\ \sigma_{moc}}$ is mainly at sub-seasonal frequencies (Figure

S2). In what follows, we focus on seasonal and decadal time scales.

|  | $< \psi'\psi'_v >$ | $< \psi'\psi'_{Z\sigma_{moc}} >$ | $< \psi'_v\psi'_{Z\sigma_{moc}} >$ |
|---|---|---|---|
| CORA | **0.41** | 0.22 | **-0.78** |
| EN4 | **0.75** | 0.13 | **-0.54** |
| GloSea5 | **0.88** | 0.21 | -0.26 |
| ECCO | **0.41** | **0.90** | -0.00 |

**Table 4: Correlation between the two leading terms of the MOC decomposition $\psi'_v$ and $\psi'_{Z\sigma_{moc}}$ and the MOC strength for the 60-month moving mean low-passed filtered time series. Correlations statistically different from zero at the 95% confidence level**
**are reported in bold.**

**4.3 Seasonality**

The MOC seasonal cycle at OVIDE is relatively consistent between the analyses (Figure 7a). The seasonal cycle of $\psi$ peaks in March, except for GloSea5 that peaks in late spring, and it troughs between July and October for CORA, EN4 and GloSea5, and November for ECCO. The peak-to-trough amplitude of the seasonal cycle varies by a factor of two between the analyses, from

8 Sv for $\psi_{CORA}$, which has the most intense seasonal cycle, to 3.7 Sv for $\psi_{ECCO}$. The seasonal cycle of $\psi'_{\sigma_{moc}}$ is similar in amplitude and phase to that of $\psi$, which confirms that the main driver of the seasonality is the seasonal variation in the depth of $\sigma_{moc}$ (Figure 7b). Velocity fluctuations $\psi'_v$ contribute more marginally to the seasonal cycle of $\psi$, with an average seasonal cycle of $\sim 2$ Sv from peak to trough (Figure 7c). Overall, the seasonal cycle of $\psi'_v$, which is at its maximum in October–November and at its minimum in July, is offset when compared to the seasonal cycle of $\psi'_{\sigma_{moc}}$. In autumn, the two seasonal cycles are opposed;

nonetheless, $\psi'_v$ shows a secondary peak in March for CORA and EN4 that amplifies the late winter peak in $\psi$ seasonal cycle. Ekman transport, which is included in $\psi'_v$, contributes to the seasonal cycle with a peak-to-trough amplitude of 1.7 Sv, with a minimum in November and a maximum in June (Figure 7a). The seasonal cycle of $\sigma_{moc}$ is very similar in all analyses, with the densest values observed in April (+0.05 kg m$^{-3}$ on average, Figure 7d) lagging the seasonal maximum of $\psi$ by one month and the least dense values in December–January (-0.06 kg m$^{-3}$ on average).


Panels a–d in Figure 8 show the variation in longitude of $\sigma_{moc}$ depth between Greenland and Portugal for the extremes of the seasonal cycle. Consistently across all analyses, $\sigma_{moc}$ is located at around 1000 m in the Iberian Basin, south-east of the NAC, rising westwards through the NAC to reach depths of ~200 m in the central Irminger Sea before deepening again in the EGC. In late winter–early spring, $\sigma_{moc}$ is denser than during the rest of the year (Figure 7d). In the central Irminger Sea and in the EGC,

upper layers densify in winter due to winter deep convection and subduction of convected water into the EGC (Piron et al., 2017; Le Bras et al., 2020). As a result, $\sigma_{moc}$, although denser in late winter, is found there at shallower depths in winter than in summer or autumn (Figures 7e,f and 8a–d). Elsewhere, and in particular in the NAC, which is the main supplier of northward transport in the MOC upper limb, $\sigma_{moc}$ is found at a greater depth at its density maximum in April than during the rest of the year (Figures 7g and 8a–d). This is because, in the NAC system, there is no seasonal variation in density at the depth of $\sigma_{moc}$ (not shown), and

the seasonal variation in $\sigma_{moc}$ density causes here a vertical shift in $\sigma_{moc}$ depth according to the average density profile, which increases downwards. In brief, the volume of the upper branch of the MOC decreases between late autumn and late winter in the EGC and central Irminger Sea, while it increases elsewhere. As in the first order, the northward transport in the upper branch of




the MOC is due to the NAC, partially offset by the southward transport in the EGC (Figure 3), this change in volume leads to an increase in the northward contribution of the NAC and a decrease in the southward contribution of the EGC to the upper MOC

transport. Figure 8e shows that, with the exception of GloSea5, it is the transport anomaly in the EGC that makes the strongest contribution to the $\psi_{\sigma_{moc}}$ seasonal amplitude anomaly, complemented by the transport anomaly in the NAC (transport anomalies in the Irminger Current and the East Reykjanes Ridge Current mostly balance each other out). Overall, the MOC is more intense in winter than in summer or late autumn. It is therefore the deep convection events triggered by intense air-sea buoyancy loss and the eddy-driven subduction of the convected water into the EGC that drive significant variation in the depth of $\sigma_{moc}$ and the

volume-driven seasonal cycle of the MOC. Seasonality in $\psi_v$ is dominated by that of the eastern boundary current, which is around 3 Sv more intense in October–November than in July–August (Figure 8f). This seasonality counterbalances that of the EGC and the recirculation to the south of the NAC, east of 20°W, which are more intense in October–November, but are directed southwards and therefore tend to weaken the MOC (Figure 8f).

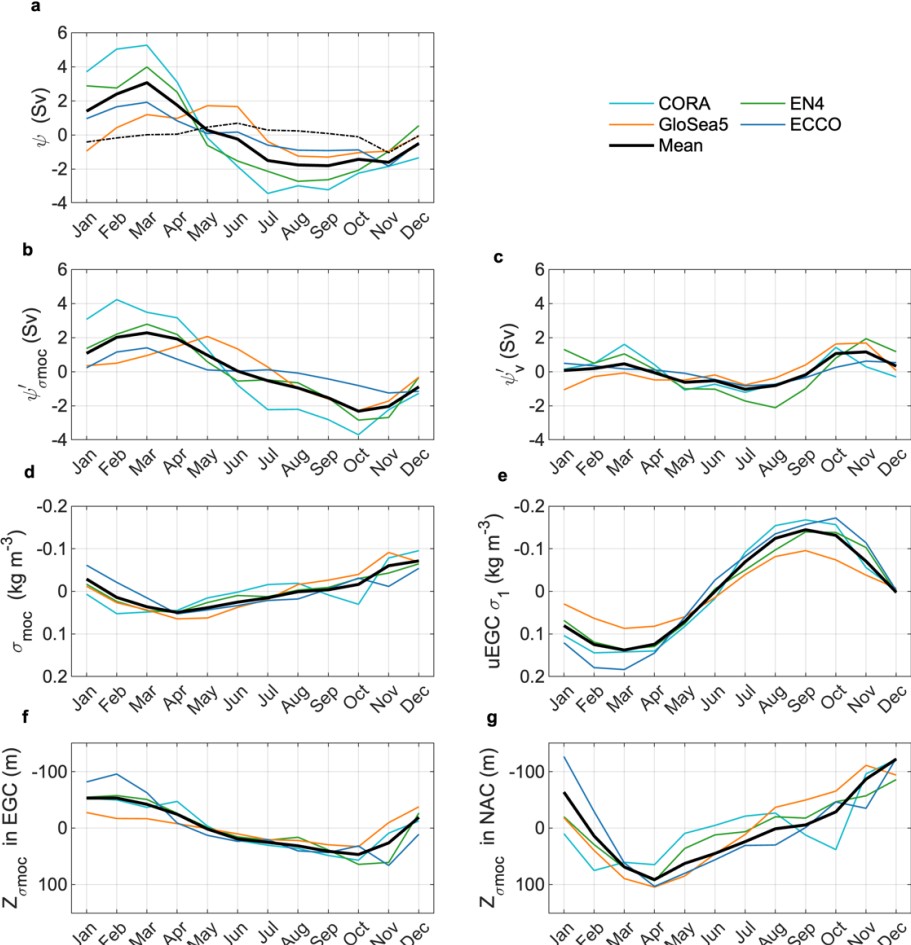

**Figure 7 : Mean intra-annual variability of (a) MOC $\psi'$ at OVIDE in Sv, (b) $\psi'_{\sigma_{moc}}$, (c) $\psi'_v$, (d) $\sigma_{moc}$ in kg m$^{-3}$, (f) $\sigma_1$ in the upper EGC (100-300 m), (e) $Z_{\sigma_{moc}}$ in the EGC defined as the southward flowing western boundary current west of 40°W (Figure 3), (f) $Z_{\sigma_{moc}}$ in the NAC defined as the broad current system east of 26°W (Figure 3). Each panel shows CORA (cyan), EN4 (green), GloSea5 (orange), ECCO (blue) and the mean of all 4 analyses. Black-dashed line in (a) is the Ekman transport, common to all analyses.**



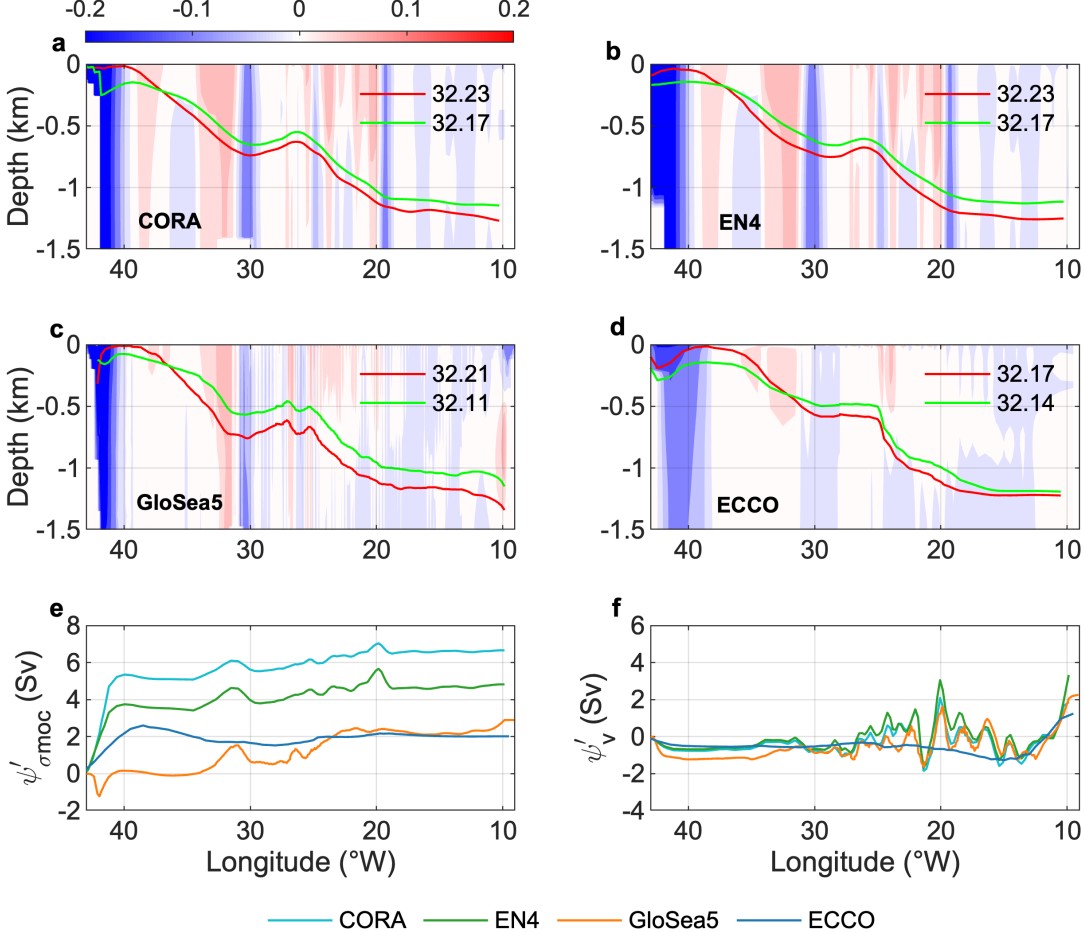

**Figure 8 :** The longitude evolution of $\sigma_{moc}$ depths averaged over February–April (red, maximum of $\psi$ seasonal cycle) and September–November (green, minimum of $\psi$ seasonal cycle) for (a) CORA, (b) EN4, (c) GloSea5 and (d) ECCO from Greenland to Portugal. The background field is the velocity perpendicular to the OVIDE line in m s⁻¹. Legend in panels a-d is density of $\sigma_{moc}$. (e) Transport anomalies corresponding to the difference between the 3-month average at the maximum and minimum of $\psi'_{\sigma_{moc}}$ seasonal cycle (February–April minus September–November) for $\psi'_{\sigma_{moc}}$. (f) Transport anomalies corresponding to the difference between the 3-month average at the maximum and minimum of $\psi'_v$ seasonal cycle (October–November minus July–August). Transport anomalies are reported in Sv and accumulated eastward from the Greenland coast in the upper limb of the MOC for $\psi'_{\sigma_{moc}}$ (e) and $\psi'_v$ (f).

### 4.4 Decadal signal

On a decadal scale, studied here from low-pass filtered time series using a 60-month moving average, CORA, EN4 and GloSea5 show that the strength of the MOC at OVIDE decreased from 1993 until 1999 for CORA and EN4 and until 2006 for GloSea5 (Figures 5 and 6, black dotted lines). After a quick recovery, the analyses do not show any particular trends during the 2010s, except for a weak relative maximum in the middle of the decade for CORA and EN4. ECCO missed the MOC decline in the 1990s, which is most often identified in analyses (see e.g. Jackson et al., 2022), but like the other analyses shows no particular trend in the 2010s. In this section we study decadal variability based on the decomposition of section 4.2 and low-passed filtered time series (Figures 5 and 6). At these time scales, $\psi'_v$ shows a positive correlation with $\psi'$, significant at the 95% confidence level for all analyses (Table 4, Figure 5). With the exception of ECCO, $\psi'_{\sigma_{moc}}$ is not significantly correlated with $\psi'$. The variability of the MOC at OVIDE on decadal timescales therefore appears mainly driven by velocity fluctuations. This is particularly true between 1993 and the mid-2000s (Figure 5). We note, however, that in all analyses, $\psi'_{\sigma_{moc}}$ decreases between



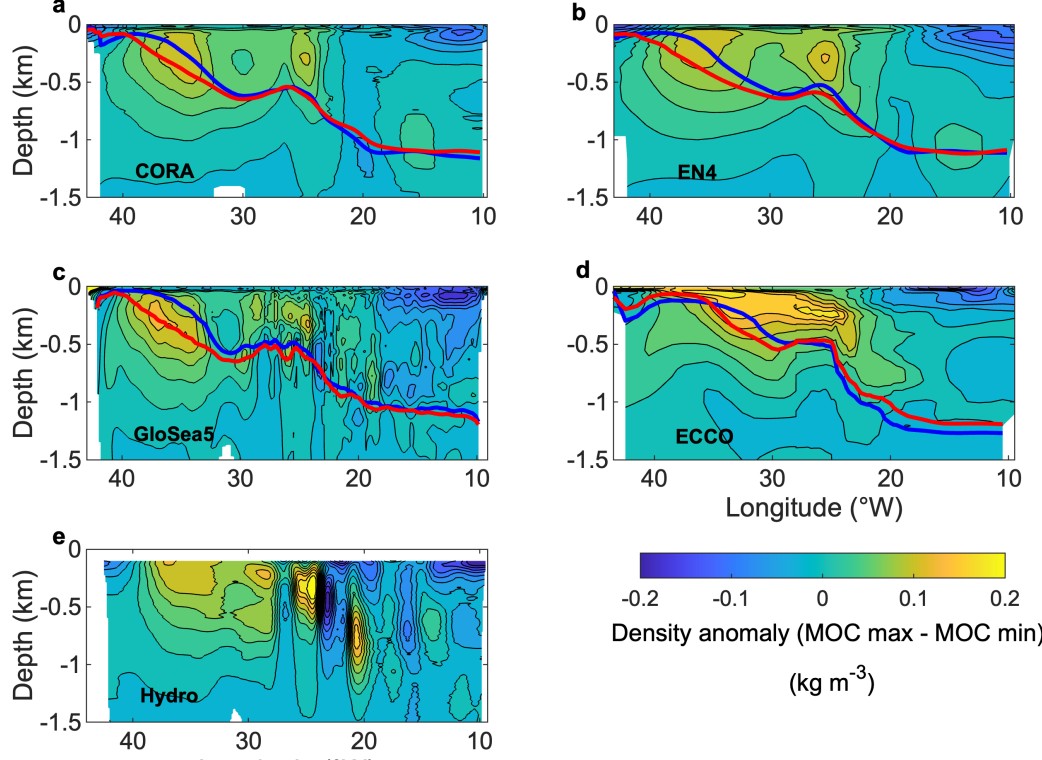

**Figure 9: Density difference (kg m⁻³) between 2015–2018 and 2004–2008 for CORA (a), EN4 (b), GloSea5 (c), ECCO (d) and OVIDE**
**hydrography (e) for 0-1500 m along the OVIDE line (Figure 1). The longitude evolution of $\sigma_{moc}$ depths averaged over 2015–2018 (blue**
**solid line) and 2004–2008 (red solid line) are superimposed for panels a - d. For the OVIDE hydrography, the density difference was**
**calculated from the cruise data and smoothed horizontally using a moving average of 70 km to remove noise due to the snapshot nature**
**of the data. Only the density difference for depths less than -0.1 km are plotted to avoid the seasonal aliasing affecting both the surface**
**layer and the $\sigma_{moc}$ depth.**

the mid-2000s and the late 2010s, while $\psi'_v$ increases over the same period (Figures 5 and 6). $\psi'_{\sigma_{moc}}$ and $\psi'_v$ show opposite

behavior and an anti-correlation computed over the entire time series lengths, significant at 95% confidence, for CORA and

EN4. To better understand this anti-correlation, we examine in Figure 9 the density anomalies along the OVIDE section for

2015-2018 (maximum of MOC) compared to 2004-2008 (minimum of MOC). In CORA and EN4 (Figure 9 a–b), the signals

show a densification broadly affecting the first 1000 meters west of the subpolar front at ~25°W, with local maxima in the center

of the Irminger gyre and east of the Reykjanes ridge, and a lightening of the upper layers east of 20°W. Similar signals are also

observed from OVIDE hydrography, which however present some eddying structures (e.g. between 20° and 25° W) due to the

snapshot nature of the measurements (Figure 9e). These density anomalies result in a change in horizontal density gradients, an

increase in geostrophic velocities east of 25°W and finally an increase in the northward transport in the upper branch of the MOC

as observed in the MOC velocity-driven component $\psi'_v$ east of 25°W (Figure 10). The densification of the upper branch of the

MOC to the west of the subpolar front contributes to a raising $\sigma_{moc}$ and results in a decrease in the volume of the upper branch

of the MOC in the Irminger current (Figure 9). Therefore, the transport due to the volume-driven MOC component $\psi'_{\sigma_{moc}}$

decreases between the two periods (Figure 10). GloSea5 shows the same structure of the upper layer density anomaly as CORA

and EN4, but weaker (Figure 9a–c), and the same transport decadal variability for $\psi'_v$ and $\psi'_{\sigma_{moc}}$ except in EGC for $\psi'_{\sigma_{moc}}$

(Figure 10). In ECCO, the positive density anomaly is intensified (Figure 9d) compared to the density anomalies in the other

analyses, and it appears subducted towards the south-east, below the subpolar front and in the thermocline, in disagreement with

the other analyses. This results in a change in the horizontal density gradients and explains the different behavior of the state



estimate (Figure 10). Overall, the variability of the MOC at OVIDE on decadal time scales results from the opposite behavior of $\psi'_v$ and $\psi'_{\sigma_{moc}}$, each of which depends on the way the analysis reproduces the anomalies in the density field. For the objective analyses CORA and EN4 and the reanalysis GloSea5, whose MOC reconstructions are in good agreement with observations

from the OVIDE line (Figure 2), the $\psi'_v$ anomaly is about twice the $\psi'_{\sigma_{moc}}$ anomaly, which explains why the entire MOC variation appears driven by velocity.

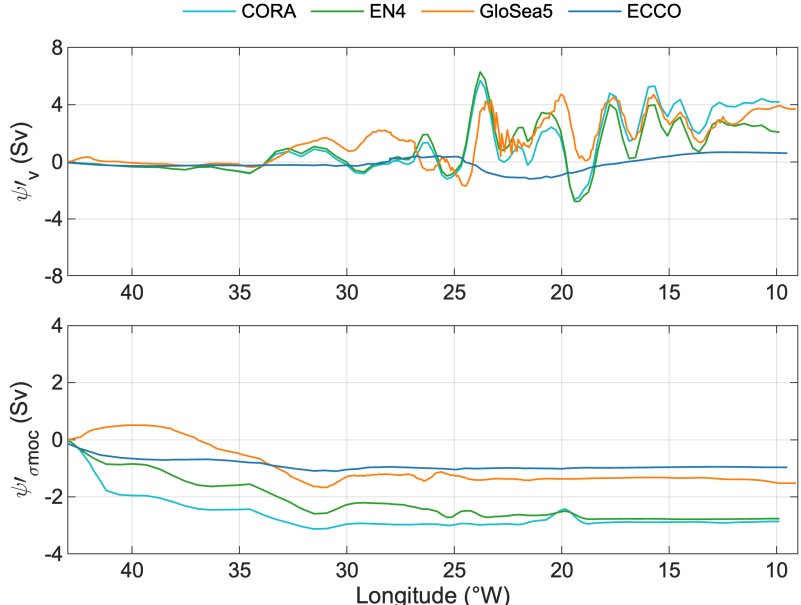

**Figure 10: Transport difference (Sv) between 2015-2018 and 2004-2008 accumulated in the MOC upper limb from the western**
**boundary eastward along the OVIDE line for $\psi'_v$ (upper panel) and $\psi'_{\sigma_{moc}}$ (lower panel). CORA is reported in cyan, EN4 in green, GloSea5 in orange and ECCO in blue.**

## 5 Discussion

The mean value of the MOC at OVIDE is between 19.7 Sv and 20.3 Sv for EN4, CORA and GloSea5 while it is 15.8 Sv and 15.3 Sv for ASTE and ECCO respectively. Estimates based on OVIDE hydrographic sections suggest that the MOC is

underestimated by ECCO, which is however the only estimator to use a four-dimensional variational estimation method ensuring dynamic consistency over the entire estimation period. Across OVIDE, the NAC transport appears decisive in determining the mean MOC strength. At 60°N, the mean value of upper limb MOC transport at OSNAP-East (Figure 1) was estimated at $17.9 \pm 0.6$ Sv for 2014-2020 (Fu et al. 2023). Note that this value was obtained by adding 1.6 Sv to the MOC lower limb transport of 16.3 Sv given by Fu et al. (2023) in order to account for a net northward transport of 1.6 Sv through OSNAP-East that occurs in

the MOC upper limb. During 2014–2020, the time-mean transport of the MOC upper limb at OVIDE (average between CORA, EN4, Glosea5) has a strength of $20.2 \pm 0.3$ Sv showing a difference of 2.3 Sv with OSNAP-East observations. This is also what is suggested by the analysis of GloSea5, which shows that the MOC at OSNAP-East is 1.2 (1.4) Sv lower than the MOC at OVIDE for the period 2014-2020 (1993-2021). The difference between the MOC at OVIDE and the MOC at OSNAP-East is most likely explained by water mass transformations in the Iceland Basin and Rockall Trough between the two sections

(Desbruyères et al., 2019). Thus, Desbruyères et al. (2019) estimated a difference of 4.2 Sv between the surface-forced transformation rates North of 45°N and North of OSNAP-East whose order of magnitude positively echoes our results. Interestingly, the MOC at OVIDE and the MOC at OSNAP-East estimated from GloSea5 show that the two series are





significantly correlated (0.83, p < 0.001) (Figure S3) suggesting that the same mechanisms drive the variability of the MOC at the two lines for the most energetic seasonal and decadal time scales.


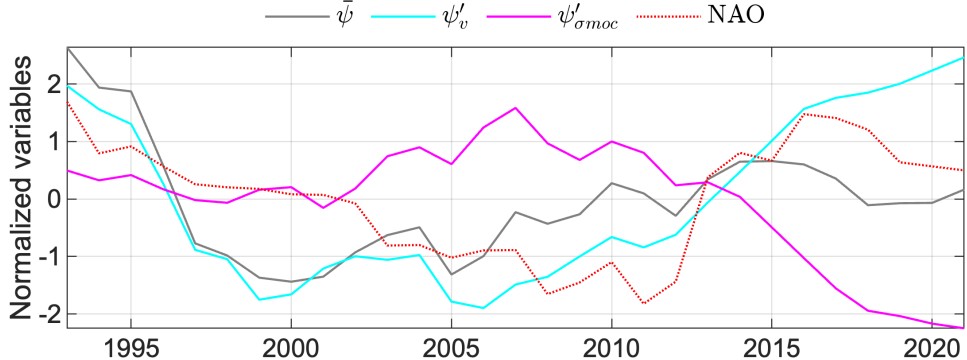

**Figure 11: Winter NAO (dotted red line), $\psi'$ (grey), $\psi'_v$ (cyan) and $\psi'_{\sigma moc}$ (magenta). Time series were normalized by their standard deviations and low-pass filtered using a moving mean with a 60-month window.**

Seasonal density changes in the upper EGC drive the seasonal cycle of the MOC. Using OSNAP data over the period 2014–2016, Le Bras et al. (2020) linked density changes in the EGC to the intermittent presence of Irminger Sea Intermediate Water (ISIW, 32.23 to 32.38 kg m$^{-3}$ in $\sigma_1$), a water mass formed offshore of the EGC and which joins this western boundary current by eddy-driven subduction. We observed that the seasonal change in density in the EGC generates a seasonal adjustment of $\sigma_{moc}$ that leads to a change in the volume of the upper limb of the MOC in the EGC and an opposite volume change in the NAC.

These volume changes result in perturbations of the MOC strength, which combine to increase the MOC strength in winter and decrease it in summer and autumn. The numerical study by Tooth et al (2023) reached the same conclusion about the importance of changes in the MOC branch volumes in explaining seasonality at OSNAP-East and identifies seasonal changes in EGC transport as a key component of seasonality. Along OVIDE, the velocity-driven seasonality is dominated by the eastern boundary current transport which opposes the volume-driven changes in late autumn. Analyzing the seasonality at OSNAP-East, Fu et al.

(2023) observed a maximum MOC strength in May and a minimum in December with a peak-to-trough amplitude of 6.2 Sv similar to the amplitudes of the CORA and EN4 seasonal cycle, which are in the high range of our estimates. The Ekman transport contributes significantly to the seasonal cycle at OSNAP-East with a peak-to-trough amplitude of 2.4 Sv between May and December, higher than the amplitude of 1.7 Sv observed between June and November at OVIDE. The difference in amplitude of the Ekman transport between OSNAP-East and OVIDE is due to differences in the orientation of the sections (Figure 1). Fu

et al. (2023) linked the seasonal variability of the MOC at OSNAP-East to that of the water mass transformation to the north of the section and to the rapid export of upper North Atlantic Deep Water (NADW, 32.23 to 32.38 kg m$^{-3}$ in $\sigma_1$) by the EGC, which causes a maximum of overturning 3-5 months after the occurrence of the transformation maximum. NADW belongs to the lower branch of the MOC (on average $\sigma_{moc}$ ~ 32.20 kg m$^{-3}$ for the period considered). The OVIDE and OSNAP-East lines follow the same path in the Irminger Sea and by linking the density variations in the EGC to volume anomalies in the MOC limbs and to

MOC strength anomalies, our results shed additional light on the results of Fu et al. (2023). Noteworthy, the seasonal cycle is controlled by the transformations of the water mass surrounding the density of $\sigma_{moc}$.

On a decadal scale, Jackson et al. (2022) conclude that there is evidence that MOC in the subpolar gyre declined from the mid-1990s to 2010 and has intensified since then. Fu et al. (2020) argue that MOC has been stable in the subpolar gyre since 1980.

Fraser et al (2021) show a stable MOC at 50°N between the mid-1990s and the mid-2000s, followed by a decrease until 2015.





Regardless of the analysis considered, we do not observe in this study any significant trends in the time series over the entire period analyzed. However a majority of analyses show a decrease in the first part of the time series, followed by a rather low MOC in 2000-2008, followed by an increase until 2018, in agreement with the conclusions of Jackson et al. (2022).

The transformation of light water masses into dense water masses in the subpolar gyre and in particular the resulting density changes in the Irminger Sea are key to the decadal-scale variability of the subpolar overturning (Figures 9 and 10). On decadal timescales, density changes in the deep convection region of the Irminger Sea lead to velocity-driven MOC changes partly offset by volume-driven changes. Interestingly, velocity-driven MOC changes are associated here with changes in NAC transport, which have already been identified in previous studies as a key element in explaining AMOC variability (Desbruyères et al.,
2013; Desbruyères et al., 2015; Kostov et al., 2023). Chafik et al. (2022) showed that the density (averaged over the first 1000 m) in the Irminger Sea was correlated with the amplitude of the MOC at OSNAP-East, while acknowledging that it had not been possible to derive a relationship between density in the Irminger Sea and atmospheric forcing because advection from the Labrador Sea was a determining factor in the variability of the density field in the Irminger Sea. We note, however, that the upper layer of the MOC is limited to depths of less than 500 m in the Irminger Sea and that this layer is ventilated every winter even during periods of moderate convection, suggesting that the impact of local forcing is a determining factor in the variability
of the MOC. Roussenov et al. (2022) identified the variability of the density field in the Irminger Sea as an indicator of MOC variability at OSNAP-East by linking a positive (negative) density anomaly in the Irminger Sea to a positive (negative) MOC strength anomaly using the Mongomery potential. In agreement with these results, we have shown in section 4.4 that the density field in the Irminger Sea acts on $\psi'_v$, increasing (decreasing) the MOC strength when the density anomaly is positive (negative).
Analyzing four years of OSNAP measurements (2014-2018), Li et al. (2019) did not find any relationship between density anomalies in the EGC and subpolar overturning. Here we show that density variations in the Irminger Sea influence the volume of the MOC on decadal time scales and hence the MOC itself via $\psi'_{\sigma_{moc}}$ (Figure 10). However on a decadal time scale, $\psi'_{\sigma_{moc}}$ and $\psi'_v$ are in opposition. In particular, during the period 2014-2018, which saw exceptional deep convection in the Irminger Sea (Piron et al., 2017), the overturning stream function anomalies $\psi'_{\sigma_{moc}}$ and $\psi'_v$ have the same order of magnitude but are of
opposite sign (Figure 10). This ultimately results in small MOC anomalies, and cancels the correlation between density anomalies in the Irminger Sea and MOC variability, showing the difficulty of understanding the variability of the MOC using an approach based solely on the analysis of density anomalies.

The North Atlantic Oscillation (NAO see Hurell, 1995) is known to be the driver of density field changes in deep convection
zones in the Labrador and Irminger Seas (Yashyaev et al., 2016; Piron et al., 2017). However, there is no one-to-one relationship. For example, Zunino et al. (2020) showed that the preconditioning of the water column by advection from the Labrador Sea allowed deep convection to continue southeast of Cape Farewell (Greenland) over the period 2014-2018 after the exceptional NAO winter of 2014 and despite forcing conditions returning to the mean. Russonov et al (2022) circumvented this difficulty by constructing composites from strong NAO events (greater than 1.6 times the standard deviation) and showed that strong NAO
events were associated with positive Irminger Sea density field anomalies and positive MOC anomalies. Our time series (Figure 11) suggest that $\psi'_v$ is correlated (r=0.74, p=0.15) on decadal scales with the NAO and that $\psi'_{\sigma_{moc}}$ is anti-correlated (r=-0.61, p=0.21). Numerous modeling studies suggest that on decadal scales the NAO precedes MOC variability (see Kim et al., 2023 for a review) but our time series are too short to confirm this statistically. In the end, the anti-correlation between $\psi'_v$ and $\psi'_{\sigma_{moc}}$ suggests that the decomposition used in this paper applied to historical climate model runs could provide more insight into the
variability of MOC and atmospheric forcing.



**6 Concluding remarks**

We studied the evolution of the MOC between Greenland and Portugal over almost three decades using four different data-driven estimators. The MOC measurements $\psi_{hydro}$ taken during the OVIDE cruises showed good agreement with the analyses where the weight of the data was the highest ($\psi_{cora}$, $\psi_{en4}$, $\psi_{glosea5}$). The state estimates $\psi_{ecco}$ deviates more from our direct
observations. OVIDE biennial observations have therefore been decisive in enabling a critical assessment of the data-driven estimators.

Although they are essentially based on the same data sets, the four analyses do show some differences, for example in the seasonal cycle of the MOC, the amplitude of which varies by a factor of two between the analyses, or the difference between
ECCO and the other analyses in the reproduction of the decadal variability of the MOC for the 1990s. However, the decomposition into velocity-driven and volume-driven components made it possible to identify variability mechanisms common to all the analyses. Thus, the seasonal variability can be ascribed to volume variations in the EGC and to transport variations at the eastern boundary. Decadal variation in MOC is driven by velocity in the 1990s. While dominated by the velocity component, decadal variation in MOC strength in the years 2005 to 2021 is damped by the volume component.

**Data availability**

OVIDE Hydrographic data are available from https://www.seanoe.org/data/00353/46448/ (Mercier et al., 2022). CORA v5.2 objectively mapped fields are available on
https://data.marine.copernicus.eu/product/INSITU_GLO_PHY_TS_OA_MY_013_052/services; EN4 objective analyses on https://www.metoffice.gov.uk/hadobs/en4/download-en4-2-2.html; GloSea5 reanalysis on
https://data.marine.copernicus.eu/product/GLOBAL_REANALYSIS_PHY_001_031/description; ECCO fields on https://podaac.jpl.nasa.gov/ECCO?tab=mission-objectives§ions=about%2Bdata; NCEP/NCAR atmospheric analysis on https://psl.noaa.gov/data/gridded/data.ncep.reanalysis.html; AVISO altimeter on http://marine.copernicus.eu/services-portfolio/access-to-products/; NAO index on
https://www.cpc.ncep.noaa.gov/products/precip/CWlink/pna/norm.nao.monthly.b5001.current.ascii.table.

**Authors' contributions**

HM and DD conceived the study. HM made the computation and wrote the first draft (PL provided the inverse model results for 2016–2021). All authors discussed the results and reviewed two successive drafts.

**Competing interests**

The authors declare that they have no conflict of interest.

**Acknowledgements**

This work received funding from the European Union's Horizon 2020 research and innovation programme under grant agreement No 862626 (EUROSEA), the French National Programme Les Enveloppes Fluides et l'Environnement (LEFE) and Ifremer. HM was supported by the Centre National de La Recherche Scientifique (CNRS). FF and AV were supported by the BOCATS2 (PID2019-104279GB-C21) project funded by MCIN/AEI/10.13039/501100011033 and together with LH, by EuroGO-SHIP
project (Horizon Europe #101094690). MF was supported by grant PTA2022-021307-I funded by



MCIN/AEI/10.13039/501100011033, by European Social Fund Plus,  and by the Portuguese Foundation for Science and Technology through projects UIDB/04326/2020 (doi:10.54499/UIDB/04326/2020), UIDP/04326/2020 (doi:10.54499/UIDP/04326/2020), LA/P/0101/2020 (doi:10.54499/LA/P/0101/2020) and CEECINST/00114/2018. We gratefully acknowledge support from the French Oceanographic Fleet to OVIDE (https://doi.org/10.18142/140). Colormaps are
from MatPlotLib Perceptually Uniform Colormaps.

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
