# Peer review of "New insights into the eastern Subpolar North Atlantic meridional overturning circulation from OVIDE"

_EGUsphere, 2024_

## Referee Comment (RC1)

**First review of "New insights into the eastern Subpolar North Atlantic meridional overturning circulation from OVIDE" by Mercier et al.**

This study is comprised of a comparison of the AMOC from Greenland to Portugal between four different "data-driven" products, ground-truthed against the long-running hydrographic section OVIDE. The authors present an interesting decomposition of AMOC variability into volume-driven and velocity-driven variability and analyze this decomposition on both seasonal and decadal time scales. They find that seasonal AMOC variability along this line is dominated by volume variability (i.e. changes in the depth of the isopycnal of maximum overturning), while decadal changes are mostly driven by velocity variability.

The manuscript is well-organized, clear, and the results are interesting and timely. The referencing and placement into the larger scientific context are appropriate. I recommend that it be accepted for publication after minor revisions, which are mainly targeted at improving the presentation of the results.

Minor comments:

L18: The authors should consider explicitly mentioning the isopycnal of maximum overturning in the abstract. It is not immediately clear to the reader who is not yet familiar with the formalism that changes in volume are due to changes in the depth of the isopycnal of maximum overturning.

L60: Recommend adding "with" after the comma.

L75: (Figure 1 caption) Consider adding the time period over which AVISO is averaged.

L118: Missing "depth" at the end of the line.

L139: The extracted dataset should be made publicly available and the citation provided in the data availability section.

L147: Typo in GloSea. Recommend rewriting "in perspective with…"

L166: It would be helpful to explicitly refer to the publication that details the OVIDE inverse method formalism.

L173: Section title and beyond: Should this be "MOC" rather than "AMOC" as it is referring to overturning strength at one latitude (expectation from L43). In general AMOC and MOC are used somewhat interchangeably in the main body of the text after a specific expectation is set up in the Introduction.

L181: referred → referenced

L236: Recommend removing "in this case" and specifying that this calculation is for the cross-correlations.

L257: Please elaborate on the standard error calculation for the seasonal cycle.

L285: (Figure 2) Can the authors make the y axis for ECCO the same as the others so that they are easier to compare?

L387: Unclear what is meant by "As in the first order".

L400: (Figure 7) It is difficult to distinguish the cyan and blue lines. This comments also applies to Figures 8 and 10.

L405: (Figure 8) Can you explain why there is a significant contribution from the eastern boundary current in Figure 8f but this is only apparent in GloSea5 in Figure 3?

L459: This is the first mention of the ASTE product (it may be left over from a previous version and should be removed).

L460: Recommend starting a new sentence after ECCO. Suggestion for starting the next sentence: "At the same time, ECCO is the only…"

L462: Please elaborate on/clarify the differences between how OVIDE and OSNAP handle the net transport across the line.

L470: Recommend removing "Thus,"

L500: "Noteworthy" is unclear here, could be replaced with "We find that" or "A new and noteworthy result is that"

L526: It is potentially also worth discussing the connection to seasonal density variations in the EGC in the context of the comparison with Li et al. 2019, not just decadal.

L534: de Jong and de Steur (2016) should also be discussed in the context of Irminger Sea convection (or at least referenced in conjunction with Irminger Sea convection somewhere).

de Jong, M. F., and L. de Steur (2016), Strong winter cooling over the Irminger Sea in winter 2014–2015, exceptional deep convection, and the emergence of anomalously low SST, *Geophys. Res. Lett.*, 43, 7106–7113, doi:10.1002/2016GL069596.

L541: It looks like the total overturning variability is also positively correlated with the NAO, not just the components. Could the authors please discuss whether this is the case?

---

## Author Comment (AC1)

We would like to thank Reviewer 1 who provided constructive comments. Reviewer 1's comments have been reproduced in black with the authors' response in blue and excerpts from the revised manuscript in italics.

**First review of "New insights into the eastern Subpolar North Atlantic meridional overturning circulation from OVIDE" by Mercier et al.**

This study is comprised of a comparison of the AMOC from Greenland to Portugal between four different "data-driven" products, ground-truthed against the long-running hydrographic section OVIDE. The authors present an interesting decomposition of AMOC variability into volume-driven and velocity-driven variability and analyze this decomposition on both seasonal and decadal time scales. They find that seasonal AMOC variability along this line is dominated by volume variability (i.e. changes in the depth of the isopycnal of maximum overturning), while decadal changes are mostly driven by velocity variability.

The manuscript is well-organized, clear, and the results are interesting and timely. The referencing and placement into the larger scientific context are appropriate. I recommend that it be accepted for publication after minor revisions, which are mainly targeted at improving the presentation of the results.

Thank you for these positive comments.

Minor comments:

L18: The authors should consider explicitly mentioning the isopycnal of maximum overturning in the abstract. It is not immediately clear to the reader who is not yet familiar with the formalism that changes in volume are due to changes in the depth of the isopycnal of maximum overturning.
Certainly. We have added the following sentence : "*We decompose the MOC strength variability into a velocity-driven component due to circulation changes and a volume-driven component due to changes in the depth of the overturning maximum isopycnal.*"

L60: Recommend adding "with" after the comma.
Done, thank you.

L75: (Figure 1 caption) Consider adding the time period over which AVISO is averaged.
Done. The sentence in the caption now reads : *"OVIDE and OSNAP-East lines plotted over the mean over 1993–2012 of the AVISO surface dynamic topography (Jousset et al., 2022)."* We also added the reference of the dataset, which was missing.

L118: Missing "depth" at the end of the line.
"depth" has been added, thank you.

L139: The extracted dataset should be made publicly available and the citation provided in the

data availability section.

We now provide a netcdf file of the time series in the Supplement.

L147: Typo in GloSea. Recommend rewriting "in perspective with…"

The typo has been corrected and the sentence simplified. The text now reads: "*The reader is referred to Jackson et al. (2016; 2019) for a discussion of North Atlantic circulation features derived from GloSea5 reanalysis and ECCO state estimate and their comparison with other analyses.*"

L166: It would be helpful to explicitly refer to the publication that details the OVIDE inverse method formalism.

Done. We have included a new sentence before the inverse model description that reads : "*The inverse model was described by Lherminier et al. (2007); the main steps of the method can be summarized as follows.*"

L173: Section title and beyond: Should this be "MOC" rather than "AMOC" as it is referring to overturning strength at one latitude (expectation from L43). In general AMOC and MOC are used somewhat interchangeably in the main body of the text after a specific expectation is set up in the Introduction.

You are right. We have replaced "AMOC" by MOC where appropriate.

L181: referred → referenced

Done.

L236: Recommend removing "in this case" and specifying that this calculation is for the cross-correlations.

Done. We deleted "in this case" and added "*for the cross-correlation r*".

L257: Please elaborate on the standard error calculation for the seasonal cycle.

Done. The sentence now reads "*The standard error was calculated as the ratio of the intra-annual standard deviation divided by the number of degrees of freedom on the assumption that MOC observations for a given month one year apart are independent.*" This standard error is now reported in section 4.3.

L285: (Figure 2) Can the authors make the y axis for ECCO the same as the others so that they are easier to compare?

Done.

L387: Unclear what is meant by "As in the first order".

We replace "As in first order" with "*As in first approximation*".

L400: (Figure 7) It is difficult to distinguish the cyan and blue lines. This comments also applies to Figures 8 and 10.

We replaced the cyan lines with ice blue lines to increase the contrast with dark blue lines.

L405: (Figure 8) Can you explain why there is a significant contribution from the eastern boundary current in Figure 8f but this is only apparent in GloSea5 in Figure 3?

Figure 8f shows the difference between autumn and summer. The eastern boundary current has a strong seasonality and is intensified in autumn compared to other seasons, hence the difference shows an eastern boundary intensification. Figure 3 shows the circulation in June-July and reveals that GloSea5 has a stronger eastern boundary current than the other analyses for this time period. To clarify this point for the reader, we have added the following comment: "*Note that although the eastern boundary current in GloSea5 was more intense in June–July than in the other analyses (Figure 3), the amplitude of its seasonality between July–August and October–November is similar to that of the other analyses.*"

L459: This is the first mention of the ASTE product (it may be left over from a previous version and should be removed).

Removed.

L460: Recommend starting a new sentence after ECCO. Suggestion for starting the next sentence: "At the same time, ECCO is the only…"

Done. Thank you.

L462: Please elaborate on/clarify the differences between how OVIDE and OSNAP handle the net transport across the line.

The point is that we determine the MOC strength by integrating the meridional overturning stream function from the surface and not from the bottom. We added a clarification: "*The net northward transport must be added to OSNAP MOC lower limb transport as it is included in MOC strengths determined by integrating the meridional overturning stream functions from the surface, as we do here.*"

L470: Recommend removing "Thus,"

Done. Thank you.

L500: "Noteworthy" is unclear here, could be replaced with "We find that" or "A new and noteworthy result is that"

Done, replaced by: "*A noteworthy result* is that …". Thank you.

L526: It is potentially also worth discussing the connection to seasonal density variations in the EGC in the context of the comparison with Li et al. 2019, not just decadal.

This point is addressed in the text from the work of Fu et al. (2023) who analyzed a longer OSNAP time series than Li et al. (2019) but, you are right, we can also cite Li et al. (2019), which we were happy to do.

L534: de Jong and de Steur (2016) should also be discussed in the context of Irminger Sea convection (or at least referenced in conjunction with Irminger Sea convection somewhere). de Jong, M. F., and L. de Steur (2016), Strong winter cooling over the Irminger Sea in winter

2014–2015, exceptional deep convection, and the emergence of anomalously low SST, *Geophys. Res. Lett.*, 43, 7106–7113, doi:10.1002/2016GL069596.

Certainly. We now make reference to de Jong and de Steur (2016). The sentence now reads : "*The North Atlantic Oscillation (NAO see Hurell, 1995) is known to be the driver of density field changes in deep convection zones in the Labrador and Irminger Seas (Yashyaev et al., 2016; Piron et al., 2017; de Jong and de Steur, 2016).*" Note that de Jong et al. (2018) was already referred to in the introduction.

L541: It looks like the total overturning variability is also positively correlated with the NAO, not just the components. Could the authors please discuss whether this is the case?

This is the case, in agreement with the fact that on a decadal scale, the variability of the strength of the MOC mainly follows that of $\psi'_v$. We have added the following sentence: "*The MOC strength is positively correlated with NAO (r=0.50; p=0.38), which is consistent with the fact that, on longer time scales, the variability of the MOC strength is mainly driven by $\psi'_v$.*"

---

## Author Comment (AC2)

We would like to thank Reviewer 2 who provided constructive comments and interesting questions. Reviewer 2's comments have been reproduced in black with the authors' response in blue and excerpts from the revised manuscript in italics.

This is an interesting analysis of the variability of the MOC expressed in sigma coordinates, separating a velocity-driven and a volume driven component. All my comments/questions are 'minor', and only provided to clarify the already very well presented analysis and results. Thank you for these positive comments.

The discussion is made along the Ovide line, but could as well have been made on more zonal sections such as OSNAP-E, or further south. The AMOC strength is calculated from the surface downward, which for the CORA and EN4 estimates gives a very strong weight to the altimetric product (the same one for the two products which might also explain partially why the time series seem rather similar between the two products despite large differences in the density field in key areas such as near the western boundary). Although this is not a core issue for this paper, I am wondering how accurate the altimetric product is for the currents close to Greenland, where one could expect larger errors in the geoid time changes, during this period of often large (at least seasonal) south Greenland ice melt (but with decadal/interannual changes), and whether this could have some impact on the reconstructed AMOC variability in some of the analyses (my guess is possibly not that much, as this is an AMOC in sigma coordinate, thus with shallow upper layer in the western part of the section). The period chosen in 1993-2021 starting thus with the advent of more precise altimetric sea level products, but before the Argo array (or its predecessors in the North Atlantic).

We agree with the reviewer's analysis. Altimetry in south-east Greenland has been assessed by Gourcuff et al (2011) for the EGC and by Coquereau and Foukal (2023) for the EGCC. These publications show that the agreement between surface currents measured by satellite altimetry and in situ observations is good. However, the assessment by Coquereau and Foukal (2023) was carried out in summer when the shelf is free of sea ice. As this is not always the case, the resolution of the seasonal cycle of the EGCC remains a challenge for both objective analyses of in situ variables and satellite altimetry. We have added a sentence to mention this point in the discussion in a paragraph dedicated to errors at the end of section 5 and repeated below after our responses to your introductory remarks.

Coquereau, A., Foukal, N.: Evaluating altimetry-derived surface currents on the south Greenland shelf with surface drifters, Ocean Sci., 19, 1393–1411, https://doi.org/10.5194/os-19-1393-2023, 2023.

Gourcuff, C., Lherminier, P., Mercier, H., and Le Traon, P. Y.: Altimetry combined with hydrography for ocean transport estimation, J. Atmos. Ocean. Technol., 28, https://doi.org/10.1175/2011JTECHO818.1, 2011.

The choice of the different products compared is relevant, and spans a wide range of approaches. In particular, I find interesting the differences between ENA4 and CORA, which basically use the same data with some objective mapping, the largest differences been in the scales retained, and what is done near bathymetry and shelves. The two products could also differ more before 1997 and the ACCE sampling in this region (and even up to 2001, before

the Argo float network became fully operational in this region). It could be interesting to mention that ECCO V4R4 as far as I know is a state estimate everywhere, except the Arctic, as well as the year of beginning and end of that run (as this is a state estimate, the adjustment is done over the whole period of the run, and meridional boundary conditions play a role). ECCO Version 4 is the first multi-decadal ECCO estimate that is truly global, including the Arctic Ocean (Forget et al., 2015). The period over which the ECCOV4r4 state estimate was constructed (1992-1997) is given in section 2.4. Following the reviewer comment, we now mention this period again at the start of the discussion (section 5).

The decomposition is interesting and suggests a difference of contributions between seasonal variability and decadal variability (different in ECCO). Interestingly, for seasonal variability (driven by the sigma change component), it seems that it is the component of the EGC ( a decrease in winter) which is the main contribution (due to denser waters there in winter, thus less volume transported southward). I am wondering how sensitive this is to the resolution of the mapped analyses. Are we sure that they do a good job separating what happens on spatial scales typically of 100 km for the seasonal time (typically for EN4 and CORA, the mapping scales are larger than this 100 km: does this play a role, in particular for EGC?). With a station spacing of a few km where the continental slope is greatest, the OVIDE hydrographic lines have better horizontal resolution in the EGC than the EN4 and CORA objective analyses. The agreement between $\psi_{hydro}$, $\psi_{cora}$ and $\psi_{en4}$ shows that the relatively low resolution of the objective analyses has a limited impact on the MOC estimate, probably because estimating transport is less sensitive to smoothing than estimating velocity. Nevertheless, the Reviewer's point is important and we now mention in the text that the spatial smoothing of the objective analyses may contribute to the difference with $\psi_{hydro}$ (paragraph at the end of section 5 reproduced below).

On decadal scales, the main result is a strong contribution of psi'v to the variability except for ECCO. However, I wonder whether its strong contribution relative to the sigma one before the early 2002 (Figure 5) could be for a period less constrained in the analyses by the density profile data, thus at least in CORA, an underestimation of the density-induced variability. The anti-correlation between the two contributions is also well explained, and expected due to the distribution of where the large density changes occur. ECCO on the other hand exhibits a rather different structure of density anomalies. For me, this clearly points to a deficiency of the state estimation of the density field, and maybe also of the circulation in ECCO (the subduction of the density anomalies in the thermocline to the southeast). We agree with the Reviewer who identifies here another source of uncertainty which is the temporal inhomogeneity of the data distribution. We now mention this point in the text (paragraph at the end of section 5 reproduced below).

At the end of section 5, we have added the following paragraph summarizing the various points made by the reviewer: "*In addition to the way in which the analyses take the observations into account (objective analysis, 3D-var assimilation, 4D-var assimilation), several factors contributing to the differences in MOC observed between the analyses can be mentioned. EN4 and CORA use the same data sets and an objective analysis but differ in their*

*choice of spatial correlation functions and therefore in the spatial scales selected. Dynamics play a more significant role in the interpolation of data by GloSea5 and ECCO, but given the different spatial resolutions of the ocean models (0.25° and 1° respectively) the way in which eddies are taken into account differs. The reconstruction of the seasonal variability of the MOC, for which the density field west of the Irminger Sea is a key parameter, is also challenged by the limitations of the data sets. While high-precision altimetry began in 1993, the Argo network was not deployed until 2002. Argo floats have little or no coverage of water depths below 1000 m and satellite altimetry will be limited for periods when the Greenland shelf is covered by sea ice. Despite these limitations, our results have highlighted the respective roles of $\psi'_v$ and $\psi'_{\sigma moc}$ in the variability of the subpolar MOC."*

Other 'minor' comments:

1. 193. As important as reporting the mean value of the mean transport would be reporting the seasonal or decadal variability of this net transport which is small for ECCO, but seems to me rather large in GloSea5. Does this net transport variability have a large impact in the decadal variability discussed later.

   First of all, we apologize for the fact that the net transport associated with GloSys5 is 1.6 Sv and not 0.8 Sv as indicated by mistake in the submission. This error has been corrected in the text. The time evolution of the net transports is now discussed, adding in the text: "*Unlike ECCO, whose net transport is relatively stable on decadal time scales, GloSea5 shows a decadal trend with an increase in net transport of ~1Sv between the end of the 2000s and the end of the 2010s.*". The increase in net transport of GloSea5 accounts for ~25% of the increase in $\psi'_v$. We speculate that the velocity changes that cause the changes in MOC strength are the same as those that cause the changes in net transport. We have added in section 4.4: "*Interestingly, the ~4 Sv increase in $\psi'_v$ appears to translate into a ~1Sv increase in net transport for GloSea5 (Figure S1)*".

2. 263: '…, the MOC strengths (-0.23, 0.23)'. I suspect these differences are estimated at the time of the Ovide sections, and then averaged. It could also be interesting to report the uncertainty on these average differences (I guess roughly (rms(MOC) with hydro)/sqrt(11)). It seems that large differences might stem from the section in 2006. Are these differences within the error bars of the psi(hydro) estimate for that year? (or is there something special that might explain that apparently larger difference with all other products (less so, for ECCO, except if one corrects for the time series mean difference)

   The reviewer is right: the differences are estimated at the time of the Ovide lines as Table 1 explains. The reader is more explicitly referred to Table 1 in the text when quoting these numbers. At the time of the 2006 survey, the geostrophic transports of the NAC and EGC estimated by inverse modelling, and hence the MOC, were relatively low (Gourcuff et al. 2011). The low EGC transport is confirmed by the analysis of an array of moorings (Daniault et al., 2010). The error on $\psi_{hydro}$ is ~2.5 Sv, but when comparing two estimators the error on the second estimator must also be taken into account. The differences $\psi_{hydro} - \psi_{cora}$ and $\psi_{hydro} - \psi_{en4}$ are ~ 3.5 Sv

for 2006, which makes $\psi_{hydro}$ compatible these two other estimates assuming an error of 1 Sv on $\psi_{cora}$ and $\psi_{en4}$, which, one'll agree, is a lower bound. The difference $\psi_{hydro} - \psi_{glosea5}$ is greater, but this was also the case for 2014 or 2021. We followed the reviewer's recommendation and added the errors on bias both in the text and Table 1.

3. 274: 'largely similar data sets…', I agree although CORA and EN4 also use Ago float data, which will have some weight, even at the time of the Ovide Cruise. Altimetry and hydrographic profiles are also used in ECCO and Glosea5, but apparently leaving much larger biases with the Hydro estimate (nicely discussed in the next paragraph). The sentence was ambiguous and rephrased as "*It should be noted that the  MOC strength estimates from the four analyses are not independent, as they are based on largely similar data sets (altimetry, Argo and ship-based hydrography).*"

4. 283: Based on Figure 2, it also seems that ECCO does not have an ERCC (which is or is not a cause for a lower NAC, and in itself would contribute the other way for AMOC)
There is a cyclonic circulation around the Reykjanes Ridge in ECCO with both a weak IC and a weak ERRC, the latter appearing to be slightly shifted eastward compared with hydrography. The two underestimates cancel each other out in the cumulative transport plot of Figure 3.

5. 291 (in Figure 1's caption) 'Biases with hydrography' should probably be 'Biases with respect to hydrography'
Corrected. Thank you.

6. 299: 'Positive velocities are directed northwards'. I thought that in Daniault et al's paper the velocities were the component normal to the section (thus usually not northward, except close to Portugal).
We meant meridional component of the current is directed northward. The sentence now reads : "*Positive velocities indicate that the meridional component of the current is directed northwards.*"

7. 459: what does ASTE refers to?
Deleted (it was left over from a previous version).

8. 497: the sentence 'NADW belongs to…'. This sentence is not very informative, and somehow does not deliver a point in the discussion? (just that it is not in what is estimated here for the upper limb?)
We agree. The sentence is neither informative nor necessary and has been deleted.

9. 518-519: 'We note, however that…'. Yes, but Chafik et al (2022) include the density anomalies up to 1000m. Although the lower part is not directly related to the anomalies of transport in the upper limb (locally), its contrast with what happens further east would be important to that depth… It could be interesting to remind what is the period investigated and on which time scales were the main results of Chafik et al (2022) established. Is the result of Chafik et al (2022) (no direct relation to local wind forcing …) relevant and why does it seem to differ from the conclusion of this paper on the role of local forcing (although not shown, just argued for…)?
We added the period investigated by Chafik et al. (2022) ie (1993–2018). We agree with the reviewer that the discussion of local versus remote forcing is only marginally

within the scope of the paragraph. The key point here is more likely the fact that the density field anomaly at the center of the Irminger Sea calculated by Chafik et al. (2022) is representative of the larger-scale density anomaly observed in Figure 9, which drives both $\psi'_v$ and $\psi'_{\sigma moc}$. We deleted the discussion about the forcing and added : "*We speculate that the Irminger Sea density anomaly calculated by Chafik et al. (2022) is representative of the density anomaly observed in Figure 9, which drives both $\psi'_v$ and $\psi'_{\sigma moc}$.*"

10. 538: maybe 'to average conditions' instead of 'to the mean'

    Changed.

11. 540 (and figure 11 caption): 'Our time series'? To which time series does this refer to (also to be mentioned in Figure 11 caption; maybe also clarify that on lines l. 523, as it is not crystal-clear which product and time series are considered; it might be 'all' (the four products), in which case, maybe do not use 'our'; also, is figure 11 an average of the four time series, or one in particular) . This might have been mentioned earlier, but the reader might lose track of it.

    We now mention in Figure 11 caption and in the text that the MOC decomposition time series are the average of CORA, EN4 and GloSea5. The text now reads: "*The MOC decomposition time series averaged of CORA, EN4 and GloSea5 (Figure 11) suggest that $\psi'_v$ is correlated (r=0.74, p=0.15) on decadal scales with the NAO and that $\psi'_{\sigma_{moc}}$ is anti-correlated (r=-0.61, p=0.21) (ECCO was not included in the average because it differs very significantly from the other estimates).*"

---

## Author Response (AR2)

Dear Editor,

We would like to thank you for the time you devoted to this manuscript. We wonder why the manuscript has been classified as a "review article" as it presents original research.

Sincerely,

Herlé Mercier